# Ni(II) Ions May Target the Entire Melatonin Biosynthesis Pathway—A Plausible Mechanism of Nickel Toxicity

**DOI:** 10.3390/molecules27175582

**Published:** 2022-08-30

**Authors:** Nina E. Wezynfeld, Arkadiusz M. Bonna, Dawid Płonka, Wojciech Bal, Tomasz Frączyk

**Affiliations:** 1Institute of Biochemistry and Biophysics, Polish Academy of Sciences, 02-106 Warsaw, Poland; 2Faculty of Chemistry, Warsaw University of Technology, 00-664 Warsaw, Poland; 3Department of Biochemistry, University of Cambridge, Cambridge CB2 1QW, UK

**Keywords:** nickel toxicity, nickel allergy, carcinogenicity, melatonin, protein damage

## Abstract

Nickel is toxic to humans. Its compounds are carcinogenic. Furthermore, nickel allergy is a severe health problem that affects approximately 10–20% of humans. The mechanism by which these conditions develop remains unclear, but it may involve the cleavage of specific proteins by nickel ions. Ni(II) ions cleave the peptide bond preceding the Ser/Thr-Xaa-His sequence. Such sequences are present in all four enzymes of the melatonin biosynthesis pathway, i.e., tryptophan 5-hydroxylase 1, aromatic-l-amino-acid decarboxylase, serotonin *N*-acetyltransferase, and acetylserotonin *O*-methyltransferase. Moreover, fragments prone to Ni(II) are exposed on surfaces of these proteins. Our results indicate that all four studied fragments undergo cleavage within tens of hours at pH 8.2 and 37 °C, corresponding with the conditions in the mitochondrial matrix. Since melatonin, a potent antioxidant and anti-inflammatory agent, is synthesized within the mitochondria of virtually all human cells, depleting its supply may be detrimental, e.g., by raising the oxidative stress level. Intriguingly, Ni(II) ions have been shown to mimic hypoxia through the stabilization of HIF-1α protein, but melatonin prevents the action of HIF-1α. Considering all this, the enzymes of the melatonin biosynthesis pathway seem to be a toxicological target for Ni(II) ions.

## 1. Introduction

Nickel enters our body by inhalation, dermal contact, and gastrointestinal ingestion [1]. Inhalation of airborne nickel nanoparticles is dangerous and potentially carcinogenic, as those particles are endocytosed and continuously release high amounts (resulting in high millimolar concentrations) of the metal ions into the cell [2]. Ni(II) ions are also released from many common-use metal tools, accessories, and jewelry upon contact with human skin. Such release elicits allergic dermatitis in approximately 10–20% of individuals [3,4,5,6]. However, cancer and nickel allergy development mechanisms have not been fully understood. One such mechanism may involve the hydrolysis of a peptide bond preceding the fragment with the Ser/Thr-Xaa-His sequence (where Xaa is any amino acid residue but proline) [7]. Due to hydrolysis, the protein activity is compromised. Factors that promote cleavage include the presence of the target fragments on flexible and surface-exposed parts of the polypeptide. The cleavage is pH-dependent, and Ni(II) ions break proteins relatively slowly at pH 7.4, but much more rapidly in locations with higher pH [7]. Interestingly, pH values in the mitochondrial matrix are high enough [8,9,10,11,12,13,14,15] to facilitate Ni(II)-assisted hydrolysis of the peptide bond.

The imidazole side chain of the histidine residue anchors Ni(II) ions. The metal ion also competes with amide protons and binds to respective nitrogens. That ultimately results in Ni(II) ion binding to four nitrogens (hence the name 4N complex): one from imidazole, one from the amide of the histidine residue, and two from amides of the preceding amino acid residues. The geometry of the final 4N complex is square planar [16]. When such a complex is formed at the Ser/Thr-Xaa-His sequence, it facilitates the acyl shift of the polypeptide preceding this fragment to the hydroxyl oxygen of the Ser/Thr residue (the first residue in the complex-forming fragment). The created ester hydrolyzes spontaneously in the second step of the overall cleavage reaction [17]. The described mechanism of Ni(II)-assisted peptide bond hydrolysis is depicted in Figure 1.

The Ser/Thr-Xaa-His sequence occurs in several human proteins. We have shown that histone H2A [18], annexins A1, A2, and A8 [19], alpha-1-antitrypsin [20], phospholipid scramblase 1, Sam68-like mammalian protein 2, and CLK3 kinase [21] contain surface-exposed fragments cleavable by Ni(II) ions. Ni(II)-prone sequences are also present in all of the melatonin biosynthesis pathway enzymes, i.e., tryptophan 5-hydroxylase 1 (TPH1), aromatic-l-amino-acid decarboxylase (AADC), serotonin *N*-acetyltransferase (SNAT), and acetylserotonin *O*-methyltransferase (ASMT). Recent reports have demonstrated that the biosynthesis of melatonin occurs in the mitochondria of virtually every human cell [22]. Therefore, the enzymes from the melatonin biosynthesis pathway (TPH1, AADC, SNAT, and ASMT) may be targets for Ni(II)-assisted peptide bond hydrolysis.

Beyond being a sleep hormone, melatonin is a potent antioxidant. It is synthesized by essentially all organisms. It is most probably evolutionarily very old, and its primary ancient function was the scavenging of free radicals [23]. One molecule of melatonin can neutralize up to ten free radicals [24]. Furthermore, melatonin is amphiphilic, allowing it to detoxify both hydrophilic compartments and lipid membranes. This makes the mitochondrial placement of the melatonin biosynthesis especially beneficial for protecting this organelle against oxidative stress. Notably, the mitochondrial concentration of melatonin was found to be at least two orders of magnitude higher than in plasma [25]. Melatonin also plays other roles, such as being an anticancer agent and an immune system regulator [26,27]. Thus, damage to the enzymes producing melatonin may have deleterious consequences, such as high levels of oxidative stress or dysregulation of the immune system. Intriguingly, melatonin modulates HIF-1α protein in normoxia/hypoxia sensing [22,28]. The same HIF-1α was shown to be affected by Ni(II) ions [29,30]. We hypothesize that this action on the HIF-1 axis is carried out by Ni(II)-assisted peptide bond cleavage of melatonin biosynthesis pathway enzymes.

Melatonin is biosynthesized from tryptophan in the following pathway. First, tryptophan is hydroxylated by TPH1. Then, the product of the reaction, 5-hydroxytryptophan, is decarboxylated by AADC, with the formation of serotonin. Next, serotonin is acetylated by SNAT, and, finally, *N*-acetylserotonin is methylated by ASMT [23,31]. Importantly, all substrates and intermediates in this pathway (tryptophan, 5-hydroxytryptophan, serotonin, and *N*-acetylserotonin) are free radical scavengers, together with melatonin, although with different specificity towards radicals [32,33].

We have started by analyzing structures of human enzymes from the melatonin synthesis pathway: tryptophan 5-hydroxylase 1 (TPH1), aromatic-l-amino-acid decarboxylase (AADC), serotonin *N*-acetyltransferase (SNAT), and acetylserotonin *O*-methyltransferase (ASMT). Then, we have selected those enzyme fragments with Ni(II)-prone sequences that are exposed on the surfaces of the proteins. Next, for these fragments, we have documented the pH-dependence of Ni(II) binding using UV-vis and CD spectroscopies, and potentiometry. Finally, using HPLC and mass spectrometry, we have measured the hydrolysis reaction rates. Overall, our work implies that melatonin biosynthesis pathway enzymes are toxicological targets for nickel ions.

## 2. Results and Discussion

Melatonin is synthesized from tryptophan in four steps by the following enzymes: tryptophan 5-hydroxylase 1 (TPH1, SP: P17752, EC 1.14.16.4), aromatic-l-amino-acid decarboxylase (AADC, SP: P20711, EC 4.1.1.28), serotonin *N*-acetyltransferase (SNAT, SP: Q16613, EC 2.3.1.87), and acetylserotonin *O*-methyltransferase (ASMT, SP: P46597, EC 2.1.1.4). There are experimentally obtained structures available for human TPH1 (e.g., PDB: 5L01), AADC (e.g., PDB: 3RBF), and ASMT (e.g., PDB: 4A6E) [34]. For human SNAT, there is a model of the structure made available by AlphaFold (AF-Q16613-F1) [35]. We found the following fragments with Ser/Thr-Xaa-His sequences in these enzymes:TPH1: ^267^EPDTCHELL^275^, **^343^HALSGHAKV^351^**,AADC: **^454^TVESAHVQR^462^**,SNAT: ^2^STQSTHPLK^10^, ^107^ESLTLHRSG^115^, **^169^ERFSFHAVG^177^**,ASMT: **^53^VRASAHGTE^61^**.

Among these fragments, we found that only those written in bold (HALSGHAKV, TVESAHVQR, ERFSFHAVG, and VRASAHGTE) are in flexible, surface-exposed fragments. This conclusion is based on the analysis of available structures, as shown in Figure 1A,C,E,G. This in silico analysis also indicated that these fragments could easily adapt their conformation during the formation of a square planar 4N hydrolytic Ni(II) complexes. The formation of these complexes was possible without any disturbance to other parts of the proteins (Figure 1B,D,F,H). Thus, we used these fragments in further studies to characterize the binding of Ni(II) ions and the cleaving of the peptide bond by this metal.

Because model organisms, like mice and rats, can be used to study nickel carcinogenicity and nickel allergy mechanisms, it seems crucial to find out whether the studied fragments are evolutionarily conserved. For this purpose, we made multiple alignments of human, mouse, rat, chicken, and frog protein sequences (Figure 2). We found that only the fragment from AADC has a hydrolytically active Ser-Xaa-His sequence in all five organisms. Furthermore, the TPH1 fragment is also relatively well conserved, with the whole nonapeptide fragment identical for humans, mice, and rats. Hydrolytically active sequences are unique, though, for respective fragments of human versions of SNAT and ASMT. Finally, all melatonin biosynthesis pathway enzymes from *Caenorhabditis elegans* and *Drosophila melanogaster* lack sequences prone to Ni(II)-assisted peptide bond hydrolysis. This analysis implies that other animals may not be good models for nickel toxicity studies, especially without respective genetic manipulations in the genes coding the enzymes of the melatonin biosynthesis pathway.

We have synthesized four nonapeptides with *N*-terminal amine and *C*-terminal carboxyl blocked by acetylation (Ac-) and amidation (-am), respectively, to prevent the interfering interaction of Ni(II) with these moieties, which are not present in the vicinity of Ser-Xaa-His sites in the native proteins. Hence, we worked on four peptides named after the source enzyme: Ac-HALSGHAKV-am (pTPH1), Ac-TVESAHVQR-am (pAADC), Ac-ERFSFHAVG-am (pSNAT), and Ac-VRASAHGTE-am (pASMT). We have previously proven that such properly selected peptides could be good models for Ni(II)-assisted cleavage of whole proteins [20].

The formation of a square planar 4N Ni(II) complex is a prerequisite for Ni(II)-assisted peptide bond cleavage. Such complexes show characteristic absorption *d-d* bands in UV-vis and CD spectra. Indeed, the studied peptides have shown the pH-dependent binding of Ni(II) ions, which was detected as the appearance of bands with the maximum at approx. 450 nm, in UV-vis spectra (Figure 3), as well as both positive and negative bands in the range of 350–650 nm, in CD spectra (Figure 4).

The detailed characterization of pH dependence of the Ni(II) complex formation is possible with potentiometry. Additionally, we used the above-mentioned data from UV-vis and CD spectroscopies to validate the model describing Ni(II) complexes speciation as a pH function for each peptide. Logarithmic protonation constants for peptides and logarithmic stability constants for Ni(II) complexes are listed in Table 1. The overlay of the data from UV-vis and CD on the speciation diagrams for all four peptides is presented in Figure 5.

Data from UV-vis and CD were consistent with the prediction of the speciation of Ni(II) 3N and 4N complexes based on potentiometric data (Figure 5). We detected only 4N complexes for the pSNAT peptide. It does not preclude the existence of 1N and 3N complexes for this peptide, but if they exist, their concentrations are too low to be confidently confirmed. All other peptides also formed 1N and 3N complexes. The peptide pTPH1 containing two histidine residues, formed an additional 2N complex, most probably by two imidazole nitrogens (N^Im^, N^Im^). 1N complexes are formed by anchoring the Ni(II) ion to the histidine imidazole nitrogen (N^Im^). 3N complexes have two more nitrogens engaged in chelation. These nitrogens are amides from histidine and the preceding residue. Finally, the binding of Ni(II) ions by imidazole and amide of histidine and two amides from two preceding amino acid residues leads to the formation of a square planar 4N complex (N^amide^, N^amide^, N^amide^, N^Im^). The latter complex is hydrolytically active. In the case of pTPH1 peptide, we can observe an additional 4N complex, with the lysine side chain deprotonated, which does not influence the hydrolytic activity of the chelate.

Comparing stability constants among different peptides (having different protonation stoichiometries) is possible after the correction for the protonation of the ligand, according to the Equation (1) [19]:(1)logK *xN=logβNiHn−xL− logβHnL

The ^*^K_xN_ is the protonation-corrected stability constant, the β_NiHn-xL_ is a Ni(II) cumulative stability constant of complexes with x nitrogen atoms coordinating metal ion, and the β_HnL_ corresponds to the protonation of the histidine residue. We calculated respective log ^*^K_xN_ values for 3N and 4N complexes (Appendix A). Although in our previous analogous analysis we concluded that low-spin 3N complexes (with square planar geometry) are more stable than high-spin 3N complexes (with octahedral geometry) [19], we do not see the confirmation of this for the peptides studied in the current work. Instead, it seems that protonation-corrected logarithmic stability constants for 3N complexes measured so far are in the narrow range of −18.82 to −20.5 (within approx. 2 log units; Appendix A). Similarly, 4N complexes for the peptides tested in this work, as well as from previous studies, show relatively uniform stabilities, with log ^*^K_4N_ ranging from −27.17 to −28.87 (Appendix A). Altogether it shows that the measured stabilities correspond well with results from the literature [17,19,36,37,38].

Beyond the stability constants mentioned above, this analysis allowed us to calculate both pH at which there is 50% of 4N hydrolytic complex formed, and the molar fractions of this species at specific pH, e.g., 7.4 and 8.2. These values are provided in Table 2. The ability of the peptide to form a 4N hydrolytic complex at specific pH is a crucial factor for the hydrolysis to occur.

In the final part of the work, we characterized Ni(II)-assisted peptide bond hydrolysis of the tested peptides. We chose physiological temperature (37 °C) and two pH values, 7.4 and 8.2, close to those existing in cytoplasm and mitochondria, respectively. Such pH values were also used in our previous publications [17,19,20,21,39]. Thus, it is also possible to compare the measured hydrolysis rates with other peptides. Using HPLC and mass spectrometry (ESI-MS) for the analysis of reaction mixtures allowed us to differentiate among substrate peptides, the intermediate product (after the acyl shift to the serine hydroxyl oxygen; Figure 1B), and products of the hydrolysis. Altogether, the quantitation of these reagents in specific time points allowed us to calculate rate constants for the two sub-stages of the cleavage, i.e., the formation of the intermediate product and the final hydrolysis. Calculated rate constants are shown in Table 3.

Progress of the cleavage reaction at pH 7.4 and 8.2 is shown in Figure 6. The amounts of intact peptides, intermediate products, and cleaved products after 24 h of reaction are depicted in Figure 7. At pH 7.4, the reaction is relatively slow, leading to minor fractions as products after 24 h. At the higher pH (8.2), closer to that found in the mitochondria, the cleavage after 24 h is much more advanced. The peptide pASMT was cleaved almost entirely during this time. Less than one-fifth of the intact substrate remained for pAADC and pTPH1. Even for the pSNAT, the peptide least prone to hydrolysis, only half of the intact peptide was found.

We observed a seeming discrepancy between the fraction of the peptides in the 4N hydrolytically active complex (Table 2) and Ni(II)-assisted peptide bond hydrolysis rates (Table 3, Figure 6 and Figure 7). Despite pSNAT forming the 4N complex the most easily, it undergoes cleavage at the slowest rate. This is in accordance with previous studies showing that bulky and hydrophobic residues (Phe in pSNAT) preceding Ser-Xaa-His fragment decrease the hydrolysis rates [17]. It is the opposite effect to that for small residues (such as glycine or alanine), where the cleavage is the fastest (as for pASMT) [17,19]. The Arg residue in proximity to metal-complexing fragment in pASMT also favors fast peptide bond hydrolysis, analogously to what was described earlier [17,19,21]. The hydrolysis rates are slowed most probably by the presence of the second His residue in pTPH1 (by competing with the formation of 4N complex) and Glu residue preceding Ser-Ala-His fragment in pAADC. The influence of Glu residue was also described previously [17,19].

Based on the above analysis, our results imply that the enzymes from the melatonin biosynthesis pathway, acetylserotonin *O*-methyltransferase in particular, are the toxicological targets for Ni(II) ions. It seems especially possible because melatonin is synthesized in mitochondria, which are known to contain pH more basic than the cytoplasm [8,9,10,11,12,13,14,15].

Importantly, Ni(II) ions induce hypoxia-like cell behavior, probably through the stabilization of HIF-1α protein. In normoxia, HIF-1α protein is not stable in the cell. The Fe-dependent prolyl hydroxylase (a cellular oxygen sensor) modifies proline residues of HIF-1α protein. Such hydroxylated prolines are recognized then by von Hippel-Lindau protein targeting HIF-1α to degradation. There is a hypothesis that Ni(II) can substitute Fe in prolyl hydroxylases, leading to the deactivation of hydroxylases. That inhibition finally leads to the earlier mentioned stabilization of HIF-1α and, consequently, to the activation of many genes responsible for the adaptation to hypoxic conditions. Heightened levels of HIF-1α protein were reported in a variety of cancers [29,30].

Hypoxia favors glycolysis, depleting pyruvate levels in mitochondria. This results in a lowering of acetyl-CoA concentration in mitochondria, negatively affecting melatonin synthesis (acetyl-CoA is a cofactor of SNAT) [22]. On the other hand, melatonin added externally restores a normal phenotype of the cell in hypoxia. Furthermore, melatonin is a direct inhibitor of HIF-1α and attenuates the expression of this transcription factor [22,26]. With the involvement of HIF-1α, melatonin protects against nickel-induced aerobic glycolysis [40]. It also alleviates nickel-induced mitochondrial dysfunction, although the main proposal for the mechanism of its action was a direct antioxidative activity [41,42]. We hypothesize here that Ni(II) induces hypoxia-like behavior through HIF-1α protein not only by substituting Fe ions in prolyl hydroxylases, but also by destroying enzymes of the melatonin biosynthesis pathway. Disturbing melatonin synthesis would turn off the natural modulator of hypoxia/normoxia status. This leads us to conclude that supplementation of melatonin may help prevent Ni(II) toxicity leading to cancer or allergy.

In conclusion, our data imply that tryptophan 5-hydroxylase 1, aromatic-l-amino-acid decarboxylase, serotonin *N*-acetyltransferase, and acetylserotonin *O*-methyltransferase, the enzymes of the melatonin biosynthesis pathway, are toxicological targets for Ni(II) ions. Cleavage of these proteins by Ni(II) ions may lower melatonin levels in mitochondria. It may result in elevated oxidative stress and induction of hypoxia-like phenotype of impacted cells. The final consequence may be carcinogenesis or allergy.

## 3. Materials and Methods

### 3.1. Materials

Nickel(II) nitrate hexahydrate, 99.999% trace metal basis, HCl, KNO_3_, HNO_3_, HEPES, and acetic anhydride were obtained from Sigma-Aldrich (St. Louis, MO, USA). *N*-*α*-*9*-Fluorenylmethyloxycarbonyl (Fmoc) amino acids, trifluoroacetic acid (TFA), piperidine, *O*-(benzotriazole-1-yl)-*N,N,N’,N’*-tetramethyluronium hexafluorophosphate (HBTU), triisopropylsilane (TIS), *N*,*N*-diisopropylethylamine (DIEA), dichloromethane (DCM), and acetonitrile were purchased from Merck (Darmstadt, Germany). TentaGel S RAM resin was obtained from Rapp Polymer GmbH (Tuebingen, Germany). The 0.1 M NaOH solution for potentiometric titrations was sourced from POCH (Gliwice, Poland) and standardized via potentiometry using potassium hydrogen phthalate (Merck, Darmstadt, Germany). Deionized, ultra-pure Milli-Q water was used for sample preparation.

### 3.2. Structural Analyses

Crystallographic structures of human TPH1, AADC, and ASMT (5L01, 3RBF, and 4A6E, respectively) and the structure model of human SNAT (AlphaFold: AF-Q16613-F1) were taken for simulations performed in BIOVIA Discovery Studio Visualizer v21.1.0.20298 (Dassault Systems Biovia Corp., San Diego, CA, USA). Ni(II) ion with a square planar geometry was incorporated into respective potential binding sites. Conformations of complexes were obtained by geometry optimization using a Dreiding-like forcefield [43].

### 3.3. Peptide Synthesis and Purification

Peptides were synthesized in the solid phase according to the Fmoc protocol using an automatic peptide synthesizer (Prelude, Protein Technology, Tucson, AZ, USA) [44]. The syntheses were performed on a TentaGel S RAM resin using HBTU as a coupling reagent in the presence of DIEA. The acetylation was carried out in 10% acetic anhydride in DCM. The cleavage was done by a mixture of 95% TFA, 2.5% TIS, and 2.5% water. Peptides were purified on the C_18_ column by HPLC (Breeze, Waters, Milford, MA, USA), recording absorbance at 220 nm and using a mix of eluting solvents A (0.1% (*v/v*) TFA in water) and B (0.1% (*v/v*) TFA in 90% (*v/v*) acetonitrile and 10% (*v/v*) water). The general purification method was a gradient of 5–45% of solvent B within 40 min, which was further modified depending on the peptide. The identity of each peptide was verified by ESI-MS (Premier, Waters, Milford, MA, USA).

### 3.4. UV-Vis and CD Spectroscopy

UV-vis spectra were recorded on the Lambda 950 UV/VIS/NIR spectrophotometer (PerkinElmer, Cambridge, MA, USA), and CD spectra on the J-815 spectropolarimeter (Jasco, Silver Spring, MD, USA), over the spectral range of 350–650 nm. Experiments with both detection methods were based on the titration of the samples containing 0.95 mM peptide and 0.9 mM Ni(NO_3_)_2_ with minute portions of concentrated NaOH in the pH range of 3.7–11.5. We used a slight excess of a peptide over Ni(II) ions to avoid precipitation of Ni(OH)_2_, which could otherwise interfere with the spectroscopic measurements. The optical path length was 1 cm in all cases.

### 3.5. Potentiometry

Potentiometric titrations were performed on a 907 Titrando Automatic Titrator (Metrohm, Herisau, Switzerland), using a Biotrode combined glass electrode (Metrohm), calibrated daily by nitric acid titrations [45]. One hundred millimolar NaOH (carbon dioxide-free) was used as a titrant. Samples (1.5 mL) were prepared by dissolving peptides in 4 mM HNO_3_/96 mM KNO_3_ to obtain 0.8–1.5 mM peptide concentrations. The Ni(II) complex formation was studied using samples in which the molar ratios of a peptide to the metal ion were 1:0.9, 1:0.45, and 1:0.3. The pH range for all potentiometric titrations was 2.7–11.6. All experiments were performed under argon at 25 °C. Three titrations were included simultaneously in calculations for protonation and five for Ni(II) complexation. The data were analyzed using the SUPERQUAD and HYPERQUAD programs [46,47].

### 3.6. Ni(II)-Assisted Peptide Bond Hydrolysis

The samples containing 0.8 mM peptide, 1.0 mM Ni(NO_3_)_2_, and 20 mM HEPES at pH 7.4 or 8.2 were incubated at 37 °C. Here, possible precipitation of a slight excess of Ni(II) ions over a peptide was not interfering with measurements because potentially forming Ni(OH)_2_ (in minute amounts, not visible to the naked eye) could dissolve in acidic conditions applied before the HPLC separation. Twenty microliter aliquots were periodically collected from the samples. The hydrolysis reaction was stopped by adding 20 µL of 2% TFA. The reagents of hydrolysis were separated on the C_18_ column by HPLC, recording absorbance at 220 nm and using the same eluting solvents and gradients as described above for peptide purification. The identities of substrate peptides, intermediate products, and final products of hydrolysis were checked by ESI-MS. Differing retention times on HPLC for substrate peptides and respective ester intermediate products allowed their distinction, as m/z values were identical for these species. The progress of the reaction was quantified from changes in the areas of reagent peaks recorded at 220 nm.

## Data Availability

The data presented in this study are available on request from the corresponding author.

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
