# Peer review of "Ni(II) Ions May Target the Entire Melatonin Biosynthesis Pathway—A Plausible Mechanism of Nickel Toxicity"

_molecules, 2022, doi:10.3390/molecules27175582_

Round 1

Reviewer 1 Report

In this paper, the authors investigate a mechanism for Ni(II) toxicity in which Ni(II) ions can break proteins involved in the melatonin biosynthesis pathway. They propose that, due to the fact that melatonin inhibits Hif-1a, which is necessary for cancer cell survival, the breakdown of melatonin the melatonin synthesis pathway could potentiate carcinogenic effects of Ni(II).

This is an interesting bioinorganic paper that should be of broad interest. There are a few minor grammar issues (e.g. line 75: "being an anticancer agent and an immune system regulator) but overall the style and grammar are very good.

I didn't have any issues with the scientific methodology or the conclusions the authors drew. However, a few details are missing in the methods section. In particular, in interest of reproducibility, more info is needed on how the authors performed HPLC (solvent gradient?). Also, the origin of the 100 mM NaOH solution used as a titrant is unclear. Was this purchased or prepared in the lab? Weighing NaOH in the lab is difficult, due to its hygroscopic nature, and NaOH solutions are typically standardized against KHP. It is not clear whether the authors did this. The source of other chemical reagents, e.g. nickel(II) nitrate, is also not provided.

Reviewer 2 Report

This paper is a one more element of a series of papers dedicated to the Nickel clivage of proteins bearing a given amino-acid sequence at their surface, (What's called ironically a slice of salami-paper")

No comment to be done about. The study is well calibrated and conducted, it follows a standard protocol already used in previous papers.

just a (very) few points (or questions):

-line 149-150 : give the reference of this work

-In Fig.3 (or line 325): Why the ratio Ni(II)/peptide is not 1:1 ? Is there any problems if working with an excess of free Nickel ?  While the reverse in sampling  experiments, line 342-343 ?

-definition of the d-d band. This is correct in UV spectroscopy but to my opinion not in dichroism's ellipticity definition. Is this denomination correct for a Ni(II)-peptide complex?

-Line 314: Don't forget to acknowledge the Protein Data Bank when citing the IDs of protein structures (in ref. list)....

I'm not a specialist in organic chemistry, but a question remains to me: what's the exact mechanism, in particular how in scheme-1 the A -> B the transposition occurs so easily (the second B-> C is ok, hydrolysis by a water molecule under pH control). Is the A->B step an equilibrium ? Is there some H/D isotopic effect or any other investigation of that kind that can reinforce it ? 
